# PAP8/pTAC6 Is Part of a Nuclear Protein Complex and Displays RNA Recognition Motifs of Viral Origin

**DOI:** 10.3390/ijms23063059

**Published:** 2022-03-11

**Authors:** Louise Chambon, François-Xavier Gillet, Maha Chieb, David Cobessi, Thomas Pfannschmidt, Robert Blanvillain

**Affiliations:** 1CNRS, CEA, INRA, IRIG-LPCV, University Grenoble-Alpes, F-38000 Grenoble, France; louise.chambon.pro@gmail.com (L.C.); francois-xavier.gillet@univ-lyon1.fr (F.-X.G.); maha.chieb@gmail.com (M.C.); t.pfannschmidt@botanik.uni-hannover.de (T.P.); 2CNRS, CEA, IBS, University Grenoble Alpes, F-38000 Grenoble, France; david.cobessi@ibs.fr

**Keywords:** *Arabidopsis*, PAP8/pTAC6, PEP/PAPs, chloroplast, biogenesis, RNA recognition motif

## Abstract

Chloroplast biogenesis depends on a complex transcriptional program involving coordinated expression of plastid and nuclear genes. In particular, photosynthesis-associated plastid genes are expressed by the plastid-encoded polymerase (PEP) that undergoes a structural rearrangement during chloroplast formation. The prokaryotic-type core enzyme is rebuilt into a larger complex by the addition of nuclear-encoded PEP-associated proteins (PAP1 to PAP12). Among the PAPs, some have been detected in the nucleus (PAP5 and PAP8), where they could serve a nuclear function required for efficient chloroplast biogenesis. Here, we detected PAP8 in a large nuclear subcomplex that may include other subunits of the plastid-encoded RNA polymerase. We have made use of PAP8 recombinant proteins in *Arabidopsis thaliana* to decouple its nucleus- and chloroplast-associated functions and found hypomorphic mutants pointing at essential amino acids. While the origin of the PAP8 gene remained elusive, we have found in its sequence a micro-homologous domain located within a large structural homology with a rhinoviral RNA-dependent RNA polymerase, highlighting potential RNA recognition motifs in PAP8. PAP8 in vitro RNA binding activity suggests that this domain is functional. Hence, we propose that the acquisition of PAPs may have occurred during evolution by different routes, including lateral gene transfer.

## 1. Introduction

Chloroplasts are of endosymbiotic origin, and as a remnant of their cyanobacterial ancestry, they have retained a small but highly conserved genome. Plastid genomes are composed of nearly 120 genes encoding housekeeping and photosynthesis genes found in multiple copies of circular DNA [1]. The components of the prokaryotic-type gene expression machinery, such as the plastid-encoded RNA polymerase PEP, ribosomal proteins, rRNAs, and tRNAs, are found among the housekeeping genes [2]. Transcription of plastid genes is due to the activity of a nuclear-encoded phage-type RNA polymerase (NEP) and the prokaryotic-type multi-subunit PEP complex. The latter transcribes the vast majority of all genes with prokaryotic-type promoters containing −35 and −10 elements, supporting the metabolic functions of plastids [3]. However, only a few plastid-encoded subunits are present in the major functional protein complexes (such as the photosystems), while most chloroplast proteins are encoded by nuclear genes and imported through the plastid envelope (reviewed in the work of [4]).

In the germinating seedling of angiosperms, ignition of chloroplast biogenesis requires light exposure (reviewed in the work of [5]), leading to a profound change in PEP activity accompanied by a complex morphological conversion of the organelle (reviewed in the work of [6]). At the transition from dark to light, a structural reorganization of the plastid-encoded polymerase (PEP) occurs when the catalytic core enzyme, composed of the α, β, β′, β″ subunits, is complexing with 12 nuclear-encoded proteins PAPs (PEP-associated proteins) [7]. Most of these proteins were previously described as part of the transcriptionally active chromosomes of the chloroplast [8]. A common feature of the PAP genes is the albino syndrome developing in homozygous mutants that carry their respective loss-of-function alleles [8]. These mutants are unable to produce a progeny and must be maintained through the harvesting of seeds from heterozygotes. The promoter activity of the PAPs is under the control of a common set of regulators with a typical profile in cotyledons exemplified with the PAP8 promoter exhibiting epidermal specificity in the dark, followed by a rapid and transient peak in the mesophyll after light exposure [9]. Although the functions associated with the PAPs produced in the epidermis remain elusive, their presence can be accountable for the detection of a fully assembled PEP complex during skotomorphogenesis in *Arabidopsis* [10]. On the other hand, as suggested by their patterns of expression, the assembly of the PAPs to the catalytic core in light would conceptually correspond to the strong production of PEP-dependent transcripts in the mesophyll cells. Therefore, the 12 PAPs play together an essential role in the early regulation of the PEP activity at the transition from skotomorphogenesis to photomorphogenesis.

According to their amino acid sequence and domain predictions, the PAPs have been classified into four functional groups (modelized by the authors of [11] and reviewed in the work of [12]): (1) PAPs related to DNA/RNA metabolism (PAP1, 2, 3, 5, 7; [8,13,14,15,16,17,18]), (2) PAPs involved in redox-dependent regulatory processes (PAP6/FLN1, its homologous FLN2, and PAP10/TrxZ; [19,20]), (3) PAPs required for protection against reactive oxygen species (PAP4/FSD3 and PAP9/FSD2; [21]) and (4) PAPs with yet unknown functions (PAP8/pTAC6, PAP11/MurE-like, PAP12/pTAC7; [22,23]). Although their molecular functions remain largely unknown, PAP8 and PAP12 could be placed between regulation and DNA/RNA metabolism of chloroplast gene transcription; in particular in the control of termination or transcriptional pausing through the binding to mitochondrial transcription termination factors (mTERFs). Indeed, PAP8/pTAC6 was found in additional recruitments at the mTERF5/MDA1-regulated transcriptional pause region of psbEFLJ [24] and as an interacting protein of GENOMES UNCOUPLED 1 (GUN1), a pentatricopeptide repeats (PPR) protein involved in retrograde signaling [25], while SL1/mTERF3 interacts with PAP12, PAP5, and PAP7 [26].

We reported that PAP8 is a protein dually localized in the nucleus and plastids with a nuclear pool that is potentially involved in the proper timing of chloroplast biogenesis. In particular, PAP8 interacts with PAP5/HMR and is essential for phytochrome-mediated signal transduction [27]. While the presence of PAP8 only in the nucleus does not restore any aspect of the mutant albinism, mutation of its nuclear localization signal (NLS) allows greening in slow motion as if the nucleus was unable to follow on chloroplast biogenesis [27]. In this respect, PAP8 has joined the growing crowd of dually localized proteins that may play an important role in the coordination of both nuclear and plastid gene expression [28,29].

Here we demonstrate that PAP8 is part of a large protein complex in the nucleus. We also demonstrate that mutations in the NLS perturb more than just PAP8 localization. Should the amino acids involved in the NLS play an important role within the PEP complex, the functions of PAP8 in the nucleus and in the chloroplast may not be so easily separated. Indeed, homology searches indicate that these residues are within a conserved region that might be involved in some RNA-mediated functions that could be relevant in the nucleus and within the chloroplast PEP complex. The closest homologous sequence to PAP8 corresponds to a rhinoviral RNA-dependent RNA polymerase. We propose that PAP8 may have appeared through lateral gene transfer during the early terrestrialization of the green lineage. Such a type of innovation, driven by viral infection, is not rare in plants. It is then possible that the ancestor of PAP8 acquired an extended 5′-end encoding a chloroplast transit peptide, following a mechanism similar to that involved in the massive transfer of cyanobacterial genes from the organelle to the nucleus.

## 2. Results

### 2.1. PAP8 Is Found in a Nuclear Complex during Photomorphogenesis

The detection of PAP8 in the nucleus and its ability to interact with PAP5 [27] prompted the question of a possible PAP8-containing larger functional complex in the nucleus. Therefore, large-scale organellar isolation was performed from 7-day-old cotyledons of Sinapis alba, followed by a blue native separation of protein complexes (BN-PAGE, Figure 1A–E).

PAP8 is detected in the heparin Sepharose chloroplastic PEP purified samples as shown in previous studies [7], but also in two discrete complexes from the nuclear protein fraction at a lower molecular weight from that of the known 1-MDa PEP complex. The lower nuclear complex (Ncpx2) may represent a subgroup of proteins that are present in the higher nuclear complex (Ncpx1) as the abundance ratio of the two complexes can differ according to different preparations (Figure 1E). Interestingly the thylakoid fraction obtained from broken chloroplasts was used as a migration marker and revealed the presence of a small but prominent PAP8-containing complex (Tcpx, Figure 1C,E). This thylakoid complex may represent a subpopulation of PAP8 that participates in the anchoring of the nucleoid to the photosynthetic membranes, as reviewed by the authors of [30]. PAP8, being associated with proteins from the larger TAC complex, may form stable interactions with those tightly anchored to the photosynthetic membranes as it is predicted for pTAC16 [31].

The detection of fairly large PAP8-containing nuclear complexes raised the question of their protein composition. Since we have identified several PAPs with predicted NLS [12] and we recently described the interaction between PAP8 and PAP5/HMR [27], other PAPs with NLS were tested as potential interactors in the nucleus using bimolecular fluorescent complementation in onion epidermal cells (BiFC, Figure 1F–J). In transient assays using GFP translational fusions, PAP8 and PAP5/HMR showed weaker nuclear signal than plastid accumulation prompting us to use a ΔcTP version of the GFP fusion to increase nuclear pools. PAP8 gave a fluorescent signal with PAP5, PAP7, and PAP12 (Figure 1F–H), and PAP7 gave a fluorescent signal with PAP5 and PAP12 (Figure 1I,J). Therefore, some proteins from the PEP complex interact together in the nucleus, likely using similar interaction domains, without excluding that these nuclear complexes may also contain unknown partners.

### 2.2. PAP8 Functional Fusions

To follow the dually localized PAP8, PAP8-GFP C-terminal translational fusions were cloned but failed to complement the pap8-1 mutant [27]. Here, a second strategy consisted in inserting the GFP between the essential cTP and the sequence of the processed PAP8 protein (cTP8-GFP-PAP8ΔcTP, Figure 2A). A new targetP search (https://services.healthtech.dtu.dk/service.php?TargetP (accessed on 28 January 2022) on the recombinant protein predicted a chloroplast transit peptide with a likelihood of 0.9941 and a processing site (PS) at position 54–55 (VVK-VD), confirming that the translational fusion did not alter its predicted localization. However, the novel algorithm consistently shortens the cTP by 5 amino acids as compared to the obsolete ChloroP algorithm (PS at position 59-60 VDDVD-AD) previously used for the cloning designs [27].

Functional complementation was tested with such GFP fusion under the control of the 1kb-*PAP8* promoter sequence (pP8) in the *pap8-1* mutant. The double heterozygous PCR-screened primary transformants (Figure 2B) were tested for the ratio of albino phenotypes in single progenies (Figure 2C). Four tested lines gave a ratio not significantly different from 1/16 albino phenotypes corresponding to the genotypic class of homozygous for *pap8-1* and azygous for the transgene “*pap8-1*/*pap8-1*; *tg*−/*tg*−” (Figure 2D,E). Hence, the recombinant transgene (genotypes “*pap8-1*/*pap8-1*; *tg*/−”: that correspond to 3/16 of the F2 progeny) restores most of the PAP8 function in the *pap8-1* mutant. A minor defect corresponding to a pale green phenotype in the center of the rosette occurred, which did not affect the life cycle of the plant or the production of seeds. The expression of the translational fusion GFP-PAP8 gave rise to similar fluorescent patterns as that of PAP8-GFP [27]. Under the viral promoter 35S, GFP was detected in the nucleoplasm and in plastids of transiently transformed onion cells (Figure 2F–I), whereas using the PAP8 promoter in planta, GFP was mainly detected in etioplast early after light exposure (Figure 2J). These results highlight that translational fusions on both sides of PAP8 gave a proper folding attested by the stability and localization of the fusion protein while only the cTP8-GFP-PAP8 retained the full molecular function. The harmlessness of GFP at the N-terminus contrasts with the deleterious effect of GFP imposing a steric hindrance at the C-terminus to the point of a total loss of functionality. We suspect a crucial physical interaction may take place at this site, likely involving the conserved W_327_F_328_ amino acids. On the other hand, the cTP8-GFP-PAP8 provides a functional framework allowing for other interesting tags or domains to be fused to PAP8.

### 2.3. Uncoupling Localization and Function of PAP8

To separate the nuclear and the plastid pools of PAP8, two recombinant genes have been cloned: a construction with a deletion of the predicted chloroplast transit peptide (ΔcTP) and a construction carrying a mutated nuclear localization signal replacing the five positively charged lysine and arginine residues by glycine (NLS^m5^) (Figure 3; [27]). Since none of these two variants could complement the *pap8-1* phenotype, a strategy consisting in reuniting both variants in the same *pap8-1* plant was developed along with a tester clone (BB647), where the NLS sequence of PAP8 was re-introduced in the permissive context (see pLC23) behind the cleavage site of the NLS^m5^ variant (Figure 3A).

Despite the presence of both variants (Figure 3B) in the double hemi-complementation test (dhc), doubly heterozygous plants (*pap8-1*/+; *tg^bb612^*/−; *tg^Ai10^*/*tg^Ai10^*) generated 1/4 of the delayed greening phenotype (Figure 3C) that did not show any improved greening compared to the NLS^m5^ line (Figure 3D). It was therefore concluded that either the PAP8 molecule needs to go through the plastid to be functional or that the mutations of the “NLS” produced a hypomorphic allele that has lost part of the molecular function required for full activity in plastids. To discriminate between these hypotheses, the BB647 clone (Figure 3A) was tested in functional complementation yielding similar results as the dhc line. The doubly heterozygous plants (*pap8-1*/+; *tg^bb647^*/−) produced a progeny of 1/4 seedling with a delayed greening (Figure 3C) that was stronger than that of NLS^m5^ (Figure 3E). Therefore, the mutations in NLS^m5^ do not solely alter the nuclear localization signal; they have a stronger impact instead. Hence, the nuclear localization and the molecular function are very likely entangled within the same stretch of functional amino acids, rendering it difficult to alter one without modifying the other.

In order to challenge these results with an independent cloning strategy as well as providing additional tools to study the PAP8 function, a recombinant PAP8 NLS^m5^ has been fused to the cTP of PAP4, the GFP, and the monopartite NLS of SV40 (Figure 4A). Again, the doubly heterozygous plants (*pap8-1*/+; *tg^LC16^*/−) produced ¼ of albino seeds showing no interference of the transgene with the phenotype of *pap8-1* (Figure 4B,C); the expression of the transgene LC16, tested with the production of fluorescence (Figure 4F), did not allow the greening of *pap8-1* (Figure 4C). Using the viral promoter 35S, transiently transformed onion cells displayed strong GFP fluorescence in the nucleus with residual amounts in plastids only detectable when the signal was saturated in the nucleus (Figure 4D,E). In stably transformed plants using the promoter of PAP8 (Figure 4F), a strong nuclear signal was observed in the epidermal cells of the cotyledon while only some weak signal was detected in the etioplast early after the end of skotomorphogenesis, as light exposure was imposed by the confocal imaging. This pattern is in agreement with previously published work on the epidermal specificity of the PAP8 promoter during dark growth [9]. In the absence of the strong NLS from SV40, the protein pool of PAP8 is much weaker in the nucleus, according to a dilution effect due to its distribution in all the small plastids from the epidermal cell. Independently restoring the capacity of the protein to go into the nucleus with the addition of a strong NLS, overcoming the activity due to the cTP, indicates that there is a competition between the localization signatures present within the protein sequence. The absence of greening of *pap8-1* carrying such a recombinant protein further indicates that the mutations on the native NLS sequence of PAP8 alter more than its capacity to travel in the nucleus.

### 2.4. PAP8 Microhomology to RDR6 Reveals RNA Binding Motifs in PAP8

Classical blastP searches on PAP8 did not reveal any protein carrying similar domains other than PAP8 orthologs. However, other similarity searches on smaller parts of PAP8 allowed the discovery of a microhomology with the rhinoviral RNA-dependent RNA polymerase RDR6 (Figure 5: µ, underlined sequence in Figure 5E and Appendix A).

The homology was extended using the structural homology searching program T-coffee-Expresso. The homologous regions were then mapped on the three-dimensional structure of RDR6, some of which being in close contact with the double-stranded RNA molecule that was used to stabilize RDR6 (Figure 5B, [32]). This prompted an experiment of RNA electromobility shift assay (rEMSA) using PAP8 with the same RNA probe rdr32 that is able to form very stable hairpins according to the RNAfold web server (Figure 5C). The electromobility of the free probe was changed by the addition of recombinant PAP8 (Figure 5D and Appendix A). The binding of PAP8 to rdr32, though, could be out-competed by the addition of tRNAs. The shifted rdr32 migrated approximately at the position where PAP8 was detected by Western blot in such native gels (Appendix A). Although the rdr32 probe is unrelated to the functional context of PAP8, the recombinant protein exhibited a marked RNA binding activity. This activity is likely not sequence specific (based on the effects of the competitors) but still supports the results from the structure prediction pointing to a function in RNA interaction. Additionally, PAP8 had an effect on the topology of the probe that was shifted from hairpins to dimers no matter whether it was boiled or not. This suggests that some unfolded regions of PAP8, as detected by NMR studies [27], may open hairpins allowing inter-molecular interactions or topology modifications. Interestingly, the structural homology region between RDR6 and PAP8 partially includes the predicted NLS. Besides the control of PAP8 localization, this domain may hold an additional molecular function associated with RNA. Then altering the NLS may also have affected this other function, producing the hypomorphic effect observed in the complementation tests.

## 3. Discussion

The dual targeting of PAPs as demonstrated for PAP5 [33] and PAP8 [27] or predicted for PAP7 and PAP12 [12], as well as other proteins such as NCP [34] and RCB [35], is apparently essential for chloroplast biogenesis. All of these proteins are linked both to the chloroplast transcriptional apparatus and to the phytochrome signaling pathway. For example, PAP5/HMR is able to interact with the phytochrome interacting factors PIF1 and PIF3 on a chromatin structure [36]. The PAP8-containing nuclear complexes and the possible interaction of the PAPs, as observed here with BiFC (Figure 1), suggest that the function of these proteins depends on their ability to interact with their partners in both the nucleus and chloroplasts. Nuclear localization signals and nucleic acid recognition motifs are both based on positively charged amino acids offering the possibility for a functional overlap, as suggested for PAP8 in this study. Moreover, these dually localized complexes may control gene expression at the chromatin level in the nucleus and in the nucleoids of chloroplasts. Hence, it is possible that PAP8 and other PAPs have been evolutionarily transferred into the chloroplast as a functional module mirroring in plastids the refined control of nuclear gene expression. In this context, PAP8-containing complexes may provide anterograde control from the nucleus building the photosynthetic apparatus and matching the plastid type to cell identity. Future work will consist in deciphering the composition of such nuclear complexes.

The presence of PAP8 in different compartments triggers questions of functionality. Does PAP8 play the same molecular function, whether it is in the nucleus or in the chloroplast, and are these compartmentalized functions dependent on each other. The uncoupling of these functions by using two distinct carriers (ΔcTP and NLS^m5^) simultaneously would have provided definitive proof only if they were totally independent. Instead, the phenotypic analysis revealed that the uncoupled nuclear function carried by ΔcTP does not provide any benefit to a plant carrying the NLS^m5^ recombinant gene. This suggests that the nuclear function of PAP8 is entirely masked by the inability of PAP8 to properly work during chloroplast biogenesis. In that context, it would be of interest to test whether uncoupling the genomes in a gun1 mutant background, for example, would affect the phenotypes of plants carrying NLS^m5^ alone or in combination with ΔcTP. The recombinant gene BB647 (cTP8-N8-NLS^m5^) is an even stronger hypomorphic allele since the 5-amino acid substitution can potentially affect both nuclear and plastid functions. Future work could test other dually localized PAPs for which the NLS removal does not affect the molecular function of the protein.

The structural homology of PAP8 with the RNA recognition motifs of the rhinoviral RNA-dependent RNA polymerase is probably more informative about its function than about its origin. The possibility for the C-terminal end of PAP8 to visit the large groove of double-stranded nucleic acid, such as what is observed with RDR6, strengthens the hypothesis of the GFP steric hindrance in PAP8 function when fused to the C-terminus. One can speculate that PAP8 could handle RNA at the exit from the PEP catalytic core, changing its topology for efficient loading of the ribosome. This function is compatible with the finding of PAP8/pTAC6 at the psbEFLJ operon with a negative impact on the pausing effect of mTERF5 [24].

Rough homology analyses performed on several PAP genes indicate that most PAPs, if not all when considering their ability to get in the chloroplast with a cTP acquisition, appeared in the same window of time during evolution [27,37]. The common ancestor of mosses in the green lineage is surely one of the first taxa that display every single component of the PEP-associated complex, including PAP8. It is then possible to speculate that, even in the absence of Rhinoviruses in their present form, a viral gene transfer occurred during the period of terrestrialization preceding the great diversification of plant organs [38]. In the meantime, PAP8 and other PAPs could have acquired a cTP following the same mechanism that permitted the massive cyanobacterial gene to transfer their signaling toward the chloroplast [39]. It remains mysterious whether a nuclear function pre-existed the plastid one and how it was maintained throughout evolution if not to offer a potent toolkit to coordinate the expression of both plastid and nuclear genomes.

## 4. Materials and Methods

### 4.1. Accessions

TAIR (http://www.arabidopsis.org/ (accessed on 28 January 2022))—*PAP8/pTAC6*: At1g21600; *PAP5/HMR/pTAC12*: At2g34640; *PAP7/pTAC14*: At4g20130; *PAP12/pTAC7*: At5g24314; *PAP10/TrxZ*: At3g06730; *EF1α*: At5g60390.

### 4.2. Live Material and Growth Conditions

*Arabidopsis thaliana*; *pap8-1*: SALK_024431 (N524431), and Col-0: SALK_6000, Ai15 (PAP8^ΔcTP^) and Ai10 (PAP8^NLSm5^) described in the work of [27]. *E. coli* DH5α strain (lacZ-ΔM15 Δ(lacZYA-argF) U169 recA1 endA1 hsdR17(rK-mK+) supE44 thi-1 gyrA96 relA1) was used for cloning. *Agrobacterium tumefaciens* strain C58C1 pMP90 was used for transgenesis. Rosetta™2 (DE3) (Novagen, MERCK, Darmstadt, Germany) cells were used for protein production with pAG21d (H6-PAP8^ΔcTP^). Mustard seeds (*Sinapis alba*) were grown for 7 days on soil under a photoperiod of 16 h of white light/8 h dark at 20 °C in a growth chamber. Cotyledons were harvested at the end of the night to avoid starch granules and immediately used for the preparation of organelles in a cold room. *Arabidopsis* plants were grown on ½-strength MS media, sucrose 1% pH_5_._7_ and 0.8% agar. Seeds were imbibed and stratified for 2 days at 4 °C, before growth at 21 °C for 3 days in darkness. Plants were then transferred to white light (30 µmol m^−2^ s^−1^).

### 4.3. Biochemical Purifications

*Sinapis alba* cotyledons (2 kg) were homogenized in ice-cold isolation buffer using a Waring blender and filtered through one layer of 56 µm mesh nylon; detailed protocol and buffers as in the work of [40]. In short, organelles were isolated by differential centrifugation followed by Percoll step (80%/40%) centrifugation, yielding the nuclear fraction at the bottom, intact chloroplasts at the step interface, and broken chloroplasts (used for thylakoids) at the top. Chloroplast fraction was submitted to heparin Sepharose batch binding, elution, then Centricon 30 kDa, giving sample “HS”. Then, the sample HS was subjected to gel filtration, elution, and Centricon 30 kDa, giving sample “GF”. The thylakoid fraction was prepared from chloroplasts resuspended in homogenization buffer containing 1% n-dodecyl β-D-maltoside for solubilization of protein complexes, passed through a 10 mL glass potter, then centrifugated 16,000× *g* for 5 min at 4 °C; the supernatant corresponded to sample “T”. Nuclear fraction 25 mL in lysis buffer; centrifugation 750× *g* for 10 min; the pellet was resuspended in 10 mL lysis, triton 0.5%, 10 min on ice plus stirring bar, then centrifugation at 750× *g* for 10 min. Pellet was resuspended in 10 mL lysis buffer; repeated as long as it was green. Resuspend the nuclei in equal pellet volume with washing buffer + glycerol to 50% (2 mL). Nuclei were sonicated with the following parameters (1 s/2 s relapse) in 1 min cycles × 3 (3 min total). Samples were centrifugated at 16,000× *g* 15 min; the supernatant corresponded to sample “N”. Nanodrop (N6× protein 2 mg/mL and DNA 270 ng/µL). Loadings for blue native gels: 50 µL (45 µL + 5 µL Coomassie blue). Run for 16 h at 4 °C; then Coomassie blue running buffer exchanged with clear running buffer for 6 additional hours. Gels were scanned on a ChemidocXRS^TM^ imaging system (BioRad, Hercules, CA, USA). Gels were transferred to a nylon membrane (BioRad), then blocked in TBS, Tween 0.1%, non-fat dry milk 5% *w*/*v*. The membrane was probed in TBS-Tween 0.1%, with the primary antibody against PAP8 [27]; secondary antibody, goat anti-rabbit conjugated with horseradish peroxidase, was used at a dilution of 1/5000. At each step, membranes were washed (5 times, 5 min in a TBS-Tween 0.1%); signal was detected using a chemiluminescent substrate (BioRad, ECL kit).

### 4.4. Cloning

Minipreps were performed using Qiagen kits and DNA in-gel purification using GeneClean III kit (MPBio, Irvine, CA, USA). PCR cloning was generated using Phusion^TM^ High-Fidelity DNA Polymerase (Thermo Scientific, Waltham, MA, USA). Final constructions were introduced in derivatives of pART27 binary vectors [41]. pBB61: alligator fragment PCR-amplified using opAt2S3_FSbflI, cctgcagggaaaccaaattaacatagggt oGFPKDEL_RBstB, ttcgaattacagctcgtccttgtacagc on pFP100 (gift of the Parcy Lab, Grenoble, France) removal of NcoI by point mutagenesis using oAliNlessF, ccatttacgaacgatagtcatggtgaagactaatc and oAliNlessR, gattagtcttcaccatgactatcgttcgtaaatgg. Alligator fragment clone SbfI, BstBI in pAi16 SbfI/BstBI [27].

### 4.5. Transformation

Electro-competent Agrobacterium were transformed with binary plasmids containing our transgenes (see Appendix A) selected on antibiotics Gentamycin, Rifampicin, and Spectinomycin for the plasmid carrying the transgene. Strains were then used for floral dip infiltration of the significant genotypes (medium: 2.2 g MS salts, 1 mL Gamborg’s 1000× B5 vitamins, 0.5% sucrose, 44 nM benzyl amino purine, 300 µL/L Silwet L-77). *pap8-1* was used as the progeny of a heterozygous plant; transgenic plants were then selected to carry the segregating mutant allele *pap8-1*, yielding albino plants in the progeny and to carry the selection marker using the corresponding antibiotic or the GFP signal in mature seeds when using the “alligator” system [42].

### 4.6. Transgenics Characterization

gDNA preparation: leaf tissues were ground in 1.5 mL reaction tubes, then homogenized in 400 µL of EB buffer (200 mM Tris HCl pH_7.5_, 250 mM NaCl, 25 mM EDTA, 0.5% SDS). After 5 min at 10,000× *g*, 400 µL of supernatant was added to 400 µL isopropanol. After 10 min at 10,000× *g*, the pellet was washed with 750 µL of EtOH 80% and then dried. DNA was then suspended in 50 µL of water. The PCR was performed with indicated primers. PAP8 (op8_rtp_F + op8i2_R) *pap8-1* (oPAP8_rtp_F + oLBb1.3) BB647 transgene (op8F257S + oP8E3_R) HPTII (ohpt1 et ohpt2). oPAP8_rtp_F, tggtggtgatggagatatcg; oPAP8_rtp_R, tttgagacactgaagtctcg; op8i2_R, aaggaagtctcagaacaacgc; oLBb1.3, attttgccgatttcggaac; oP8E3_R, tagtcactcattgcacatcg; EF1α: F, caggctgattgtgctgttcttatcat; R, cttgtagacatcctgaagtggaaga. oHPT1, ttcgatgtaggagggcgtgg; oHPT2, ggtcaagaccaatgcggagc.

### 4.7. Statistical Analysis

Percentages were compared using ε-test; statistical values were confronted to the table of normal distribution (Fisher Yates: statistical tables for biological, agricultural, and medical research (Oliver and Boyd, Edinburgh, U.K.) with α set to 0.05 or to retrieve *p*-values.

### 4.8. Protein Sub-Cellular Localization

Transient expression in onion cells (bulb sliced to ~16 cm^2^) was conducted using the Biolistic PDS 1000/He Particle Delivery System (BioRad) (1100 psi, 10 cm traveling distance) with DNA onto 1 µm gold particles (Seashell Technology^TM^, San-diego, CA, USA) following instructions. After 16 to 40 h in the dark at 24 °C, the epidermis was peeled and observed by fluorescence microscopy with a Nikon AxioScope equipped with FITC filters and an AxioCam MRc camera. Pictures acquired with Nikon’s Zen system. Confocal microscopy was performed on a Leica TCS SP2 or a Zeiss LSM800 using standard settings. Protein localization of stably transformed plants was examined on cotyledons.

### 4.9. rEMSA

Polyacrylamide gel 5%: acrylamide/bisacrylamide (29:1) 40%, TBE 1×, APS 10%, Temed 1%. Running buffer: TBE 10×: Tris base, boric acid, EDTA pH_8_. Binding mix: rdr32 (probe) 100 nM, buffer first-strand 5×, tRNA (0.2 mg/mL), DTT 1 mM, glycerol 2%, RNasin (10 U/µL), PAP8 (from 0 to 8 µM). Polyacrylamide gel was polymerized with APS and Temed at room temperature. Wells were washed with a syringe filled with running buffer. Pre-run at 4 °C in running buffer for 1 h at 90 volts. Meanwhile, the binding mix was prepared on ice. The probe is denatured using heat (95 °C for 30 s) and then directly stored on ice to keep it single-stranded (with a size equivalent to the hairpin). The run took about 75 min, at 90 volts, before in-gel cy5 detection using Amersham^TM^ ImageQuant (Chicago, IL, USA).

### 4.10. Protein Production

Rosetta2 cells were grown overnight in 50 mL LB with 100 µg/mL of carbenicillin and 34 µg/mL of chloramphenicol at 37 °C. A total of 1 L of LB + antibiotics was then inoculated and cultivated at 37 °C to 0.1 OD600. At 0.6 OD600, the temperature was decreased to 16 °C, and 0.5 mM of isopropyl β-D-1-thiogalactopyranoside was added. After an overnight induction, cells were harvested by centrifugation at 5500× *g*, for 25 min, at 4 °C. The cell pellet was resuspended and sonicated in 30 mL of lysis buffer: 50 mM Tris HCl, pH_8_, 0.5 M NaCl, 10 mM β-mercaptoethanol, 20 mM imidazole pH_8_ with a Complete^TM^ Protease inhibitor Cocktail tablet (Roche, Basel, Switzerland). The lysate was centrifuged at 15,000× *g*, for 40 min, at 4 °C. The purification was performed at 20 °C. After filtration, the supernatant was applied onto a NiNTA column in 50 mM Tris HCl, pH_8_, 0.5 M NaCl, 20 mM imidazole pH_8_. After elution, PAP8 was loaded on a Superdex 200 10/30 and eluted with 10 mM Tris HCl pH_8_, 50 mM NaCl, 5 mM DTT.

## Figures and Tables

**Figure 1 ijms-23-03059-f001:**
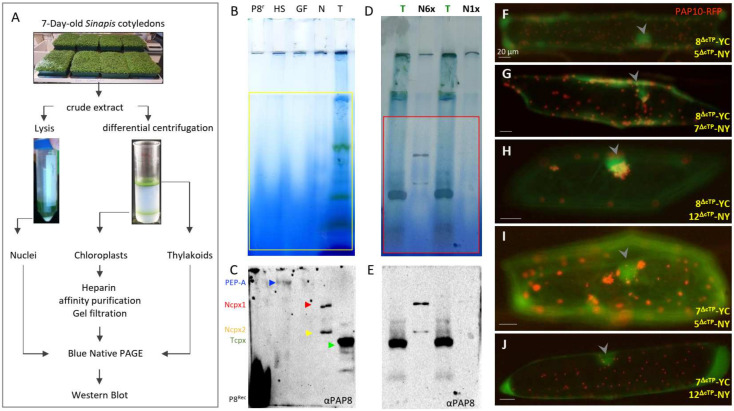
PAP8 is detected within a nuclear subcomplex. (**A**) Organelle fractionation, purification scheme, and sample processing. BN-PAGE, blue native polyacrylamide gel electrophoresis. (**B**,**D**) BN-PAGE and corresponding Western blot analyses in (**C**,**E**) using the PAP8 antibody. P8r, recombinant PAP8 protein purified from *E. coli*; HS heparin Sepharose fraction produced with the intact chloroplast sample; GF, gel filtration sample following HS; N, N6×, and N1× different loading of the sonicated and soluble nuclear fraction (see Section 4). T, thylakoid fraction from broken CP. (**C**) arrowheads, blue for PEP-A, red for the Ncpx1 (large nuclear complex), yellow Ncpx2 (smaller complex), green for the thylakoid PAP8-containing complex, and αPAP8 for the primary anti-PAP8 antibody. (**F**–**J**) Bimolecular fluorescence complementation tests using in combination PAP8^ΔcTP^-YC (8^ΔcTP^-YC) with PAP5^ΔcTP^-NY (5^ΔcTP^-NY) in (**F**); 8^ΔcTP^-YC with PAP7^ΔcTP^-NY (7ΔcTP-NY) in (**G**); 8^ΔcTP^-YC with PAP12^ΔcTP^-NY (12^ΔcTP^-NY) in (**H**); 7^ΔcTP^-YC 5^ΔcTP^-NY) in (**I**) and 7^ΔcTP^-YC with (12^ΔcTP^-NY) in (**J**); PAP10-RFP was used as internal positive control for transfection. Arrowheads indicate nuclei. Transgenes expressed under CaMV35S promoter (see [27] for published control experiments). Scale bars equal 20 µm.

**Figure 2 ijms-23-03059-f002:**
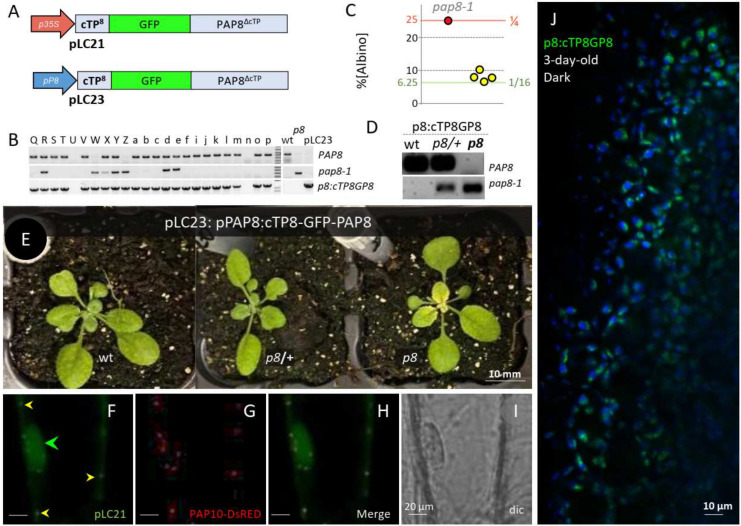
A PAP8 N-terminal translational fusion is functional. (**A**) Structure of the recombinant DNA for the production of GFP translational fusions p35S, CaMV35S promoter; pP8, 1-kb PAP8 promoter; cTP8, predicted chloroplast transit peptide from PAP8. (**B**) Sample DNA gels for the PCR-based genotyping of the primary transformants (T1) according to [27]; wild-type PAP8 allele, (op8_rtp_F + op8i2R); mutant allele pap8-1, (op8_rtp_F + oLBb1.3) and transgene p8:cTP8GP8, (oGFP_Fmfe + oP8E3_R). Wt, wild-type; p8, pap8-1 genomic DNA and pLC23, plasmid DNA containing the transgene. The doubly heterozygous plants have all three bands. (**C**) Dot plot of the percentage of albinos obtained in the segregating offspring from the doubly heterozygous plants as independent T1 (yellow spots). Functional complementation corresponds to a ratio of 1/16 (6.25%). ε-test, α = 0.05; Fisher Yates: the 4 tested lines Z, ɣ, η, R in yellow circles gave a ratio not significantly different from 1/16, n[albino]/Ntotal: 27/353; 44/706; 22/284; 28/352, respectively (εZ = 0.989; εɣ = 0.019; εη = 0.943; εR = 1.181; all ε < 𝒰5% = 1.96). (**D**) Single T2 plant genotyping, primers as in (**B**); all plants positive for the transgene (p8::cTP8GP8, pPAP8::cTP8-GFP-PAP8), the status is given for the PAP8 locus as follow: wt, PAP8,PAP8, p8/+, PAP8/pap8-1 and p8, pap8-1/pap8-1. (**E**) Three-week-old plants corresponding to representative genotypes given in (**D**). (**F**–**I**) Transiently expressed pLC21 (p35S::cTP8-GFP-PAP8ΔcTP) in onion epidermal cells; dual localization in the nucleus (green arrowhead) and in plastids (yellow arrowheads); PAP10-RFP was used as internal positive control for transfection. (**J**) Confocal imaging on *Arabidopsis* cotyledons stably expressing pP8::cTP8-GFP-PAP8ΔcTP early after light exposure of skotomorphogenetic seedlings; the picture is a merge of two channels: GFP in green: protochlorophyllide in blue.

**Figure 3 ijms-23-03059-f003:**
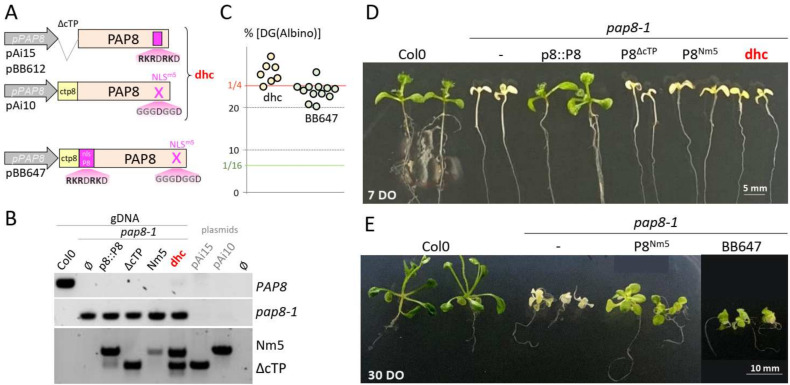
The region with the predicted NLS provides functions in addition to the nuclear localization signal. (**A**) Schematic illustration of the different PAP8 recombinant genes. cTP, chloroplast transit peptide; NLS, nuclear localization signal; m5, five substitutions leading to amino acid changes of K and R in G; dhc, double hemi-complementation test. (**B**) PCR-based genotyping of the different lines showing in particular that dhc possess the two constructions (Nm5 and ΔcTP). (**C**) Dot plot of the percentage of the delayed greening phenotype obtained in the segregating offspring from the doubly heterozygous plants from dhc (light orange dots) or independent T1 for BB647 (light green dots). A ratio of 1/4 (25%) corresponds to absence of genetic interaction between the transgene and the *pap8-1* allele: no complementation. (ε-test, α = 0.05; Fisher Yates) dhc #04: 44/148 (ε = 1.26); #19: 115/543 (ε = 2.179); #22: 63/234 (ε = 0.663); #33: 83/292 (ε = 1.297); #36: 44/149 (ε = 1.212); #40: 15/47 (ε = 1.017); #46: 98/373 (ε = 0.558) sum of all is 462 [Albino] over 1324 [WT] ε_somme_ = 0.838 < 𝒰_5%_ = 1.96. BB647 #Na: 79/279 (ε = 1.229); #Za: 91/365(ε = 0.03); #Ka: 53/235 (ε = 0.897); #Ab: 91/366 (ε = 0.06); #Ec: 73/314 (ε = 0.734); #Lb: 84/340 (ε = 0.125); #Ob: 65/279 (ε = 0.672); #Ub: 43/212 (ε = 1.708); #Vb: 55/229 (ε = 0.348); #Gc: 56/269 (ε = 1.689); #41: 65/266 (ε = 0.214); sum: 755/3154 (ε_somme_ = 1.397 < 𝒰_5%_ = 1.96). (**D**) 7-day-old (7 DO) or (**E**) 30-day-old (30 DO) phenotype of the given genotypes.

**Figure 4 ijms-23-03059-f004:**
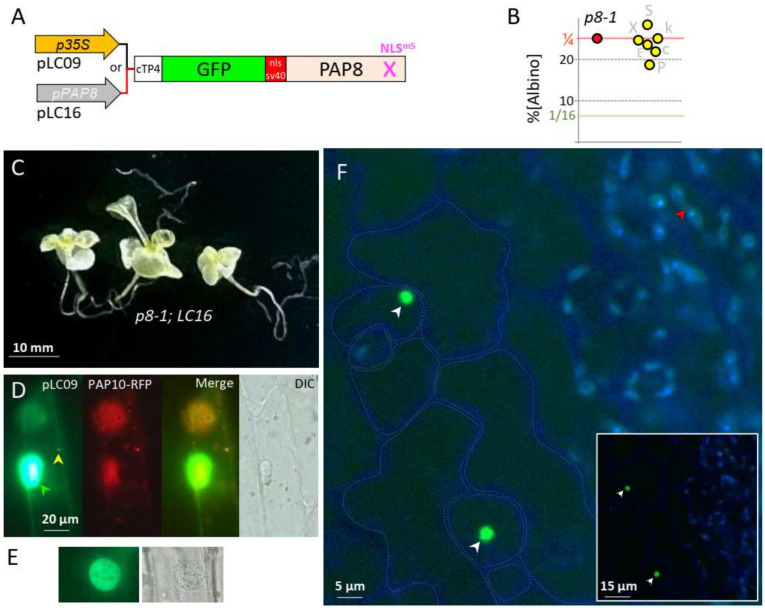
SV40 NLS fused to PAP8 provides a robust nuclear epidermal marker. (**A**) Schematic illustration of the PAP8 recombinant gene: cTP4-GFP-NLS^SV40^-PAP8^NLSm5^; cTP4 corresponds to the chloroplast transit peptide of PAP4 and NLSSV40, the NLS from the simian virus 40 (SV40). The transgene is either under the control of CaMV35S promoter in pLC09 or the PAP8 promoter in pLC16. (**B**) Dot plot of the percentage of albinos obtained in the segregating offspring from the doubly heterozygous-independent T1 for LC16 (yellow dots). A ratio of 1/4 (25%) corresponds to absence of genetic interaction between LC16 and the *pap8-1* allele: no complementation. LC16 #E: 137/566 (ε = 0.441); #X: 201/790 (ε = 0.285); #P: 118/616 (ε = 3.685); #k: 138/536 (ε = 0.395); #S: 199/682 (ε = 2.4); #c: 152/667 (ε = 1.361); sum: 945/3857 (ε_somme_ = 0.720 < 𝒰5% = 1.96). (**C**) 30-day-old phenotype of the genotype *pap8-1*, *pap8-1*; LC16. (**D**) Transiently expressed pLC09 (p35S::cTP8-GFP-PAP8^ΔcTP^) in onion epidermal cells; dual localization in the nucleus (green arrowhead) and in plastids (yellow arrowheads); PAP10-RFP was used as internal positive control for transfection. DIC, differential interference contrast image. (**E**) Non-saturated image of a nucleus and its corresponding DIC image. (**F**) Confocal imaging of *Arabidopsis* cotyledons stably expressing pP8::cTP4-GFP-NLS^SV40^-PAP8^ΔcTP^ during skotomorphogenesis; the picture is a merge of two channels: GFP in green: protochlorophyllide in blue; the blue channel has been electronically saturated to draw the cell contours in the epidermal layer; inset is the original image. White arrows show nuclei; the red arrow show reduced GFP signal.

**Figure 5 ijms-23-03059-f005:**
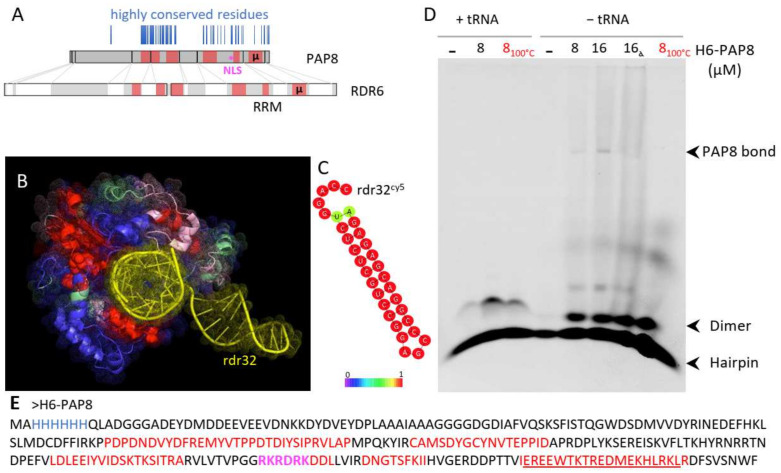
PAP8 microhomology to RDR6 reveals RNA binding domains. (**A**) Domain mapping on the amino acid sequence of PAP8 according to the structural homology comparison with RDR6 (pdb:4K50) using T-coffee expresso. Highly conserved residues in the PAP8 orthologous family of proteins are represented as vertical blue lines; µ, amino acid microhomology between PAP8 and RDR6; RRM, RNA recognition motifs in RDR6. (**B**) Amino acid painting of the three-dimensional fold of RDR6 according to the homology with PAP8: red, good; pink, average; green, bad; blue, absent in PAP8 and yellow, double-stranded RNA sequence named rdr32 hereafter. (**C**) RNAfold server prediction for the topology of rdr32 RNA sequence used as a cy5-marked probe. (**D**) RNA electromobility shift assay rEMSA using rdr32cy5 as RNA probe and H6-PAP8 produced in *E. coli*; 8_100 °C_, boiled PAP8 protein prior to setting the interaction assay with the probe; 16α, 16 µM of PAP8 + PAP8 antibody (5% of 10,000×). (**E**) Recombinant H6-PAP8 amino acid sequence; histidine tag in blue; homology regions with RDR6 in red; microhomology underlined and NLS in magenta.

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
