# Peer review of "PAP8/pTAC6 Is Part of a Nuclear Protein Complex and Displays RNA Recognition Motifs of Viral Origin"

_ijms, 2022, doi:10.3390/ijms23063059_

Round 1
Reviewer 1 Report
The work “PAP8/pTAC6 is part of a nuclear protein complex and displays RNA recognition motifs of viral origin” submitted to IJMS as research article by Chambon and co-workers presents the characterization of PAP8 localization and potential role in binding RNA. PAP8 is part of a group of proteins associated with the Plastid Encoded RNA Polymerase that are essential for chloroplast and plant development. Besides, it’s has been reported to be dual plastid and nuclear localised, joining a reduced number of PAPs that take part in light and phytochrome signalling and chloroplast biogenesis. In this work the authors studied PAP8 in more detail and found that PAP8 is found in nuclear complexes interacting with other PAPs, analyse a series of fusion proteins to untangle the function vs localisation of PAP8, and suggest a potential origin and molecular function binding RNA. Overall, the experiments are well done, presented, and discussed. After reading the manuscript I only have a few comments and suggestions for the authors:
Line 45-48. During the transition from dark to light, PEP changes not only its structure but activity. Recently a fully assembled complex had been detected in dark as mentioned a few lines below (Ref 10). Light induces an activation of PEP that could depend on other factors and not only structural reorganization or formation of the complex, and this should be stated also in the introduction.
Line 55: It would be better to specify that this promoter activity is happening in cotyledons.
Line 60: This is a hypothesis and should be written as such. The assembly of PAPs with the PEP core hasn’t been reported to be cell or tissue specific, although PAP promoters suggest that possibility.
Line 106 - Figure 1. There are some abbreviations in the legend for 1A that are not shown in the figure, and I suggest adding them to Fig 1A. Also, the abbreviations need to be consistent (e.g. I, intact CP in line 108 vs iCP in line 111). Fig1B-E: Would be better with marks for the molecular weights.
Line 181: Do the four lines correspond to primary transformants genotyped in Figure 2B? the letters don’t match and is confusing. I suggest that the names/letters of the lines are removed from line 181.
Line 191 and line 176: Contradictory information. Please clarify for Figure 2J in legend and / or text. Are these etiolated cotyledons (line 176 “during skotomorphogenesis”, figure 2J “dark”) or de-etiolating (line 191 “early after light exposure”)?
Line 252 and line 277: Contradictory information as in figure 2J. Needs clarification. If the seedlings are etiolated, the process is skotomorphogenesis, “early after skotomorphogenesis” as in line 252 suggest light exposure and hence transitions into photomorphogenesis.
Line 228: Legend figure 3B. The description of the genotyping figure seems to be incorrect as the samples mentioned (#R, #Y…) are not in the gel image lines.
Line 327: Maybe it’s worthy to mention here the other PAPs that interact with PAP8 in the nucleus. To my knowledge PAP7 and PAP12 haven’t been located in the nucleus before, although it has been described both have NLS motifs.
Line 333: do the authors mean “control of gene expression in the nucleus”?
Line 332: I would suggest a comment here about the overlap of the PAPs with NLS and with a potential RNA/DNA function that would support the hypothesis.
Line 452: Remove Gene expression from subtitle as the section only describes protein localization
Line 459: Setting details for confocal microscopy.
Line 460: Not sure hypocotyls are presented anywhere in this work. Correct this sentence or text/figures as needed.
References are needed: line 50 (PAP mutants are albino); line 60 (PEP transcription activity in light), line 83 ref 27 (Liebers 2020).
Minor issues and typos:
Line 119: Transgenes
Line 139: Transient assays with the whole protein not shown.
Line 171: Three
Line 180-181: Figure 2B and Figure 2C not 1B and 1C.
Line 189: ref to Figure 2A for construct with 35S.
Line 192: Figure 2J, not 1J.
Line 208: pLC23 and Figure 3A
Line 263: marker
Line 276: imaging of
Line 319: Recombinant H6-PAP8 instead of (H6-PAP8).
Line 390: Photoperiod?
Line 392: S. alba cotyledons
Line 421: “and” derivatives
Line 433: pap8-1 italics
Author Response
[IJMS] Manuscript ID: ijms-1600905 - Minor Revisions
March 09th, 2022
Response to Reviewer 1: (in blue)
The work “PAP8/pTAC6 is part of a nuclear protein complex and displays RNA recognition motifs of viral origin” submitted to IJMS as research article by Chambon and co-workers presents the characterization of PAP8 localization and potential role in binding RNA. PAP8 is part of a group of proteins associated with the Plastid Encoded RNA Polymerase that are essential for chloroplast and plant development. Besides, it’s has been reported to be dual plastid and nuclear localised, joining a reduced number of PAPs that take part in light and phytochrome signalling and chloroplast biogenesis. In this work the authors studied PAP8 in more detail and found that PAP8 is found in nuclear complexes interacting with other PAPs, analyse a series of fusion proteins to untangle the function vs localisation of PAP8, and suggest a potential origin and molecular function binding RNA. Overall, the experiments are well done, presented, and discussed. After reading the manuscript I only have a few comments and suggestions for the authors:
Response to Reviewer 1: The authors thank the Reviewer for the careful reading of the manuscript and the helpful suggestions to improve it.
Line 45-48. During the transition from dark to light, PEP changes not only its structure but activity. Recently a fully assembled complex had been detected in dark as mentioned a few lines below (Ref 10). Light induces an activation of PEP that could depend on other factors and not only structural reorganization or formation of the complex, and this should be stated also in the introduction.
R: We agree with the reviewer therefore we rephrased the introductory sentence to : “ light exposure … leading to a profound change in PEP activity accompanied with a complex morphological conversion of the organelle”.
Line 55: It would be better to specify that this promoter activity is happening in cotyledons.
R: This information has been added to the sentence in line 55.
Line 60: This is a hypothesis and should be written as such. The assembly of PAPs with the PEP core hasn’t been reported to be cell or tissue specific, although PAP promoters suggest that possibility.
R: Following reviewer’s suggestion we have reformulated the sentence as: “On the other hand, as suggested by their patterns of expression, the assembly of the PAPs to the catalytic core in light, would conceptually corresponds to the strong production of PEP-dependent transcripts in the mesophyll cells.”
Line 106 - Figure 1. There are some abbreviations in the legend for 1A that are not shown in the figure, and I suggest adding them to Fig 1A. Also, the abbreviations need to be consistent (e.g. I, intact CP in line 108 vs iCP in line 111). Fig1B-E: Would be better with marks for the molecular weights.
R: Thanks to the reviewer for noticing, we have removed the unnecessary abbreviations in the legend and homogenized the text accordingly. Marking molecular weights on native gels would be inaccurate: we used a thylakoid fraction with visible super-complexes as a reference for gel separation. The PAP8 signal in these lanes rendered our natural marker an interesting sample as well.
Line 181: Do the four lines correspond to primary transformants genotyped in Figure 2B? the letters don’t match and is confusing. I suggest that the names/letters of the lines are removed from line 181.
R: The lines correspond to those in figure 2C; only R and Z are presented in the genotyping gels in Fig.2B. The other two lines (Greek letters) come from a second round of genotyping with similar results that are not presented in 2B. The names have been removed in line 181.
Line 191 and line 176: Contradictory information. Please clarify for Figure 2J in legend and / or text. Are these etiolated cotyledons (line 176 “during skotomorphogenesis”, figure 2J “dark”) or de-etiolating (line 191 “early after light exposure”)?
R: We rephrased the legend: “early after light exposure of skotomorphogenetic seedlings”
Line 252 and line 277: Contradictory information as in figure 2J. Needs clarification. If the seedlings are etiolated, the process is skotomorphogenesis, “early after skotomorphogenesis” as in line 252 suggest light exposure and hence transitions into photomorphogenesis.
R: The sentence has been clarified to take into account the technical constrain of confocal imaging: “ … some weak signal was detected in the etioplast early after the end of skotomorphogenesis, as light exposure was imposed by the confocal imaging.”
Line 228: Legend figure 3B. The description of the genotyping figure seems to be incorrect as the samples mentioned (#R, #Y…) are not in the gel image lines.
R: the legend has been corrected.
Line 327: Maybe it’s worthy to mention here the other PAPs that interact with PAP8 in the nucleus. To my knowledge PAP7 and PAP12 haven’t been located in the nucleus before, although it has been described both have NLS motifs.
R: The reviewer is right, we have previously described all potential dually-localized PAPs to which PAP7 and PAP12 belong (ref. 12). The information is added in the beginning of the paragraph.
Line 333: do the authors mean “control of gene expression in the nucleus”?
R: yes we rephrased to make it clearer: “ …mirroring in plastids the refined control of nuclear gene expression”.
Line 332: I would suggest a comment here about the overlap of the PAPs with NLS and with a potential RNA/DNA function that would support the hypothesis.
R: Comment added as: “Nuclear localization signals and nucleic acid recognition motifs are both based on positively charged amino acids offering the possibility for a functional overlap, as suggested for PAP8 in this study.”
Line 452: Remove Gene expression from subtitle as the section only describes protein localization
R: removed.
Line 459: Setting details for confocal microscopy.
R: standard settings were used for image acquisition.
Line 460: Not sure hypocotyls are presented anywhere in this work. Correct this sentence or text/figures as needed.
R: Hypocotyl top cells are often observed for GFP fluorescence with much less auto-fluorescent background. There is no such data in the figures so we removed “hypocotyl” from the sentence.
References are needed: line 50 (PAP mutants are albino); line 60 (PEP transcription activity in light), line 83 ref 27 (Liebers 2020).
R: References have been added [8] for “PAP mutants are albino”. Sentence line 60 has been changed and [27] for line 83.
Minor issues and typos:
Line 119: Transgenes. (Corrected)
Line 139: Transient assays with the whole protein not shown. (Corrected)
Line 171: Three (Corrected)
Line 180-181: Figure 2B and Figure 2C not 1B and 1C. (Corrected)
Line 189: ref to Figure 2A for construct with 35S. (reference added)
Line 192: Figure 2J, not 1J. (Corrected)
Line 208: pLC23 and Figure 3A (Corrected)
Line 263: marker (Corrected)
Line 276: imaging of (Corrected)
Line 319: Recombinant H6-PAP8 instead of (H6-PAP8). (Corrected)
Line 390: Photoperiod? (yes: long day photoperiod)
Line 392: S. alba cotyledons (Corrected as Sinapis alba cotyledons)
Line 421: “and” derivatives (sentence corrected as: “PCR-cloning were generated using PhusionTM High-Fidelity DNA Polymerase (Thermo Scientific). Final constructions were introduced in derivatives of pART27 binary vectors”.
Line 433: pap8-1 italics (Corrected)
Robert Blanvillain
Reviewer 2 Report
This paper is the continuum of the story presented by the authors in the paper "Nucleo-plastidic PAP8/pTAC6 couples chloroplast formation with photomorphogenesis". The author confirmed that PAP8 is part of different complexes whether it locates in the nucleus or in the chloroplast with an additional technique: BN PAGE. They report the results of their functional fusions, localisation-uncoupling assay and RNA binding assay that would be surely crucial for the ongoing molecular and biochemistry work on PAP8 complexes.
The paper is clearly structured and well written, the experiments are properly conducted. Therefor I would recommend it for publication.
Minor comment:
- Please include "Arabidopsis thaliana" in the title and/or the Abstract
- Sentence lane 96-98: I hardly understand that very long sentence; would you reformulate?
- Lane 245 "no interaction of the transgene with the phenotype": I fear that there is a shortcut here since a gene cannot interact with a phenotype; could you reformulate.
Author Response
[IJMS] Manuscript ID: ijms-1600905 - Minor Revisions
March 09th, 2022
Response to Reviewer 2: (in blue)
This paper is the continuum of the story presented by the authors in the paper "Nucleo-plastidic PAP8/pTAC6 couples chloroplast formation with photomorphogenesis". The author confirmed that PAP8 is part of different complexes whether it locates in the nucleus or in the chloroplast with an additional technique: BN PAGE. They report the results of their functional fusions, localisation-uncoupling assay and RNA binding assay that would be surely crucial for the ongoing molecular and biochemistry work on PAP8 complexes.
The paper is clearly structured and well written, the experiments are properly conducted. Therefor I would recommend it for publication.
Response to Reviewer 2: The authors thank the Reviewer for the kind comments on the manuscript.
Minor comment:
- Please include "Arabidopsis thaliana" in the title and/or the Abstract
R: Arabidopsis thaliana was added to the abstract
- Sentence lane 96-98: I hardly understand that very long sentence; would you reformulate?
R: We rephrased as: “Such type of innovation, driven by viral infection, is not rare in plants. It is then possible that the ancestor of PAP8 acquired an extended 5’-end encoding a chloroplast transit peptide, following a mechanism similar to that involved in the massive transfer of cyanobacterial genes from the organelle to the nucleus.”
- Lane 245 "no interaction of the transgene with the phenotype": I fear that there is a shortcut here since a gene cannot interact with a phenotype; could you reformulate.
R: we meant “interference”: sentence is corrected.
Robert Blanvillain